behaviour/physiology

circadian rhythm, behaviour, social, mice, sex-based differences

**Author for correspondence:**
Johanna H. Meijer
e-mail: j.h.meijer@lumc.nl

# Group housing and social dominance hierarchy affect circadian activity patterns in mice

Yuri Robbers, Mayke M. H. Tersteeg, Johanna H. Meijer and Claudia P. Coomans

Laboratory for Neurophysiology, Department of Cell and Chemical Biology, Leiden University Medical Center, Leiden, The Netherlands

  YR, 0000-0001-8887-5079; JHM, 0000-0001-6619-5312; CPC, 0000-0003-0883-458X

In this study, we investigated the effect of social environment on circadian patterns in activity by group housing either six male or six female mice together in a cage, under regular light–dark cycles. Based on the interactions among the animals, the social dominance rank of individual mice was quantitatively established by calculating Elo ratings. Our results indicated that, during our experiment, the social dominance hierarchy was rapidly established, stable yet complex, often showing more than one dominant mouse and several subordinate mice. Moreover, we found that especially dominant male mice, but not female mice, displayed a significantly higher fraction of their activity during daytime. This resulted in reduced rhythm amplitude in dominant males. After division into separate cages, male mice showed an enhancement of their 24 h rhythm, due to lower daytime activity. Recordings of several physiological parameters showed no evidence for reduced health as a potential consequence of reduced rhythm amplitude. For female mice, transfer to individual housing did not affect their daily activity pattern. We conclude that 24 h rhythms under light–dark cycles are influenced by the social environment in males but not in females, and lead to a decrement in behavioural rhythm amplitude that is larger in dominant mice.

Analyses at the 'group' level of circadian organization will likely generate a more complex, but ultimately more comprehensive, view of clocks and rhythms and their contribution to fitness in nature.—Schwartz and co-workers [1]

# 1. Introduction

In a natural environment, the light–dark cycle is the most important determinant of the daily (circadian) rhythm. However, light-independent factors can also affect the circadian rhythm, including temperature [2], behavioural activity [3–5] and time-restricted feeding [6]. Social cues can entrain circadian patterns in group members when they are housed under constant environmental conditions [7–10]. Here, we examined whether the presence of social cues can affect the temporal activity patterns in mice housed in regular light–dark cycles and, if so, whether the social dominance hierarchy plays a role in this.

Mice are gregarious and social animals who form a complex organization within their group [11]. Living in a group has benefits as well as drawbacks, and there are many factors that influence the socioecology of a population of animals, determining—among others—whether the individuals would be more egalitarian or more despotic, with a dominance hierarchy [12–14]. When housed in a group, male mice tend to form a social dominance hierarchy [15–17], in which one or more dominant mice— the 'alpha mice'—have priority over the subordinate mice with respect to food and females [18,19]. Dominance hierarchies tie in with but do not necessarily equate to social networks that are based on agonistic, grooming and sniffing interactions [17]. More recently, it has been realized that all-female groups of mice also form dominance hierarchies [20,21]. These social dominance hierarchies can be identified by observing characteristic interactions between the mice, including chasing, fighting and biting behaviour, as well as appeasement and avoidance behaviour [15]. Socioecological constraints determine the ideal group size for any social species [12], and changing numbers of individuals within a given spatio-temporal niche can have a strong effect on the social dynamics, possibly changing the dominance hierarchy, lowering social cohesiveness or even necessitating the leaving of the social group by some individuals [16,22].

Since subordinate mice often have reduced access to food due to the presence of dominant nest-mates [15,19], and less aggressive mice (a common character trait of the less dominant mice) are more capable of exploring new spatial niches [14], we hypothesized that subordinate mice also are more capable of exploring new temporal niches. Since mice are normally nocturnal, we hypothesize, therefore, that the subordinate mice would display more daytime activity, thereby occupying a temporal niche that is less dominated by other mice, the so-called Fujimoto effect [23]. This hypothesis, based on older studies showing that wild Norway rats or white mice co-adapt in place with their conspecifics, but with temporal segregation between the low- and high-ranking individuals [23–25], has not been tested at a circadian timescale, i.e. activity patterns with a period of approximately 24 h, such as the sleep–wake cycle. In nature, there would not just be benefits, such as stress avoidance [26] or easier access to food resources [27], but also costs to the switch of temporal niche such as exposure to predators [28], paired with less or no social protection from predators [29]. All these findings are from studies of male mice, and the few data available on female mice suggest that their interactions and the cost–benefit analysis is far more subtle, and possibly more complex [30]. The hypothesis of a temporal niche shift when more individuals are present would be consistent with a previous observation that mice in the wild become more active during the day in response to an increase in population density [27] and the finding that social dominance may influence sleep patterns and quality in group-housed mice [10]. By contrast, it is possible that the behavioural activity of animals is enhanced by simultaneous activity of conspecifics [31]. In order to study harmonization, animals are typically placed in a constant environment, rendering differences in free-running period, and accordingly relative synchronization can be studied. The latter approach has been followed in the field of circadian research [7,8].

To investigate the effect of social hierarchy on behavioural activity rhythms under light–dark cycles, eight groups consisting of six same-sex mice were housed together, and each animal's circadian activity and position in the dominance hierarchy was measured using a wireless transponder and video recordings, respectively. The mice were then transferred to individual housing, and their circadian activity was again measured. In contrast with our hypothesis, we found that dominant male mice had increased daytime activity compared with subordinate mice when group housed, but not when housed individually. In addition, we observed a hierarchal ranking among the female mice, but this hierarchy was not associated with differences in daytime activity. We hypothesized that dependent on the hierarchy of the animal, the mice would perceive either group housing or individual housing as stressful, which in turn could affect daytime activity. This perceived stress could be reflected in changes in body weight of the animals as well as in increased levels of the stress hormone corticosterone [32]. To test this, we measured body weight as well as corticosterone in our experiments as an indication of perceived

stress and observed that corticosterone was not correlated with the level of dominance or daytime activity. In line with this, body weight stopped increasing when animals were singly housed, at least for male mice. Finally, we found that the corticosterone levels in male mice decreased after they were transferred to individual housing, whereas no change was observed in female mice.

# 2. Material and methods

## 2.1. Animals

All animal experiments were approved by the institutional ethics committee on animal care and experimentation at Leiden University Medical Center, Leiden, The Netherlands. For this study, a series of eight replications of the experiment were performed. In four of these replications, a group of six adult eight-week-old male C57BL/6JOlaHsd mice were group housed in a $425 \times 265 \times 180$ mm type III-h cage; in the other four replications, a group of six eight-week-old female C57Bl/6JOlaHSD mice were group housed. Mice are weaned at 21 days of age, sexually mature at 35 days of age and are considered adult from eight weeks onwards [11,33], though they continue to gain weight until they are at least six months of age or, especially with some breeds including C57Bl/6 J mice, longer [34]. Each cage consisted of a main compartment containing 16 g of nesting material (tissue paper) and a food compartment with standard laboratory rodent chow and water available ad libitum (see electronic supplementary material, figure S1). The mice were housed under a 12 h/12 h light–dark regime at 21–22°C and approximately 50% humidity. All mice were labelled using a shaving mark (see electronic supplementary material, figure S2) and, under isoflurane, fitted with a subcutaneous Trovan ID100 UNIQUE radio frequency identification (RFID) transponder, providing each mouse with a unique ID; the transponder was detected by a sensor at the gate between the main compartment and the food compartment. All mice were checked daily for their welfare and weighed weekly by taking them from the home cage and placing them on a digital weighing scale (grams in two decimals).

Using the implanted transponder, each time a mouse crossed the gate between the two compartments, an event was recorded and all events were graphically represented in an actogram. Total diurnal, nocturnal and overall activity were determined, and the ratio of diurnal activity to nocturnal activity could be calculated for each mouse. These four measures were used in order to test whether differences in activity between dominant and submissive mice have a circadian component, and—if so—when activity is different.

The behaviour of the mice was also recorded using an Axis 221 network camera equipped with a wide-angle lens and analysed in order to determine the social dominance hierarchy among the mice. Because the C57BL/6 J strain is relatively non-aggressive [15], biting was not suitable as a measure of dominance. Instead, we monitored fighting and chasing behaviour, threat displays, mounting behaviour, avoidance and appeasement, in accordance with other studies determining dominance in mice [15,20,21,35–37]. Each sequence of such behaviours was considered an interaction; for each agonistic interaction, the losing (i.e. subordinate) mouse was considered to be the mouse that left the site of the interaction first or—in the case of chasing behaviour—the mouse that was being chased. In cases in which the outcome was difficult to interpret (for example, if the view was partially obscured by nesting material), the interaction was noted but excluded from the analysis. The analysis was performed using data collected in the first 3.5 days of the experiment.

After the three- to six-week group housing period, each group of mice were separated and placed in six clean new cages, with each cage having the same configuration as the previous cage. The mice were then monitored for an additional two to four weeks, using the RFID transponders, and their behaviour was analysed as described above. Supporting data are available at the Dryad Digital Repository [38].

## 2.2. Plasma corticosterone levels

Blood samples were obtained via a small tail incision at two different times: 1 h after the lights were turned on at zeitgeber time (ZT)1 and 1 h before the lights were turned off (ZT11), at which plasma corticosterone levels are typically at lowest and highest, respectively [39]. The whole blood samples were collected in capillary tubes, which were centrifuged at 4°C. The plasma fraction was collected, and total plasma corticosterone concentration (in ng ml$^{-1}$) was measured using the Corticosterone High Sensitivity EIA immunoassay from ImmunoDiagnostic Systems (East Boldon, UK) [40,41].

## 2.3. Elo rating

The dominance rank of each individual within its group was determined using Elo ratings [42] with the parameter $k$ derived through maximum-likelihood estimation [43]. As the mice were littermates and had been housed together since birth, the dominance hierarchy was assumed to have been established before the start of the observations. Based on tests of the method using the first 12 days of three pilot experiments, the dominance hierarchy within the group could be reliably established within the first approximately 40 h, meaning that no position changes occurred anymore after these approximately 40 h. Thus, adding a safety margin, we used the data collected within the first 3.5 days (84 h) of each replication of the experiment as a burn-in period to determine each animal's rank within the hierarchy. Elo ratings are considered a reliable way of determining dominance within a hierarchy, as it can capture complex social dynamics and has the added advantage of providing an analysis of the hierarchy's stability [42,44,45]. With the Elo rating system, each mouse's rating is determined by the outcome of interactions with other mice; the mouse being rated receives rating points whenever it wins an interaction. The value of the rating points gained from winning an interaction is proportional to the likelihood that the interaction would be won, which is determined by the opposing mouse's rating. In other words, the higher the opponent's rating, the more rating points the mouse receives upon winning the interaction. Similarly, when the mouse loses an interaction, that mouse's rating points are reduced. Consequently, a higher rating corresponds with a higher degree of dominance. Based on the analysis by Vilette *et al.* [43], we chose to calculate the standard Elo rating with a starting value of 1000 for each mouse, since we had no prior information about the dominance rank and the value for $k$ derived using maximum-likelihood estimation, optimizing within the 0–500 range. To confirm the stability of the final dominance ranking, a stability index, $S_t$, was calculated for the group of mice [44,45]; an $S_t$ value greater than 0.81 is considered to indicate high stability [45].

## 2.4. Detrended fluctuation analysis

Detrended fluctuation analysis (DFA) is a commonly used method for quantifying non-stationary physiological time series [46] and was used to analyse the activity patterns of the mice. DFA $\alpha$-values closer to 1 are generally associated with healthier animals than values further removed from 1 [47,48].

## 2.5. Statistical analysis

Activity patterns were analysed visually using double-plotted actograms. Because the data do not comply with the requirements for parametric testing (i.e. they are neither normally distributed nor sufficiently similar in variance), comparisons between groups were made using the Mann–Whitney $U$-test for independent samples [49], and the Wilcoxon signed-rank test for dependent samples [50]. Correlations were calculated using Kendall's $\tau_b$ rank correlation coefficient, as this version of Kendall's rank correlation analysis has been specifically designed to deal well with ties in the ranks, and is also suited to smaller datasets [51]. When applicable, adjustments for small sample sizes [52] were used. All statistical tests, graphs and Elo calculations were performed or created using R [53] and several of its subsidiary packages. Daytime, night-time and total activity rates were measured continuously, and, where applicable, an average activity rate per hour was used.

# 3. Results

## 3.1. Group housing: dominance scores

For each replication, the dominance hierarchy was established by basing Elo ratings with maximum-likelihood estimated $k$ (48–112, when optimizing in the 10–500 range) on the agonistic interactions that occurred during the first 3.5 days of analysis (this involved 342, 126, 120 and 132 interactions for the four replications with males, and 294, 103, 114 and 138 interactions for the four replications with females). Overall, the dominance hierarchy was very stable with $S_t$ values ranging from 0.9381 to 0.9667 for the male mice and 0.9177 to 0.9423 for the female mice. An $S_t$ value greater than 0.81 is considered to indicate high stability [45]. Thus, the social dominance hierarchy was stable among both male and female groups.

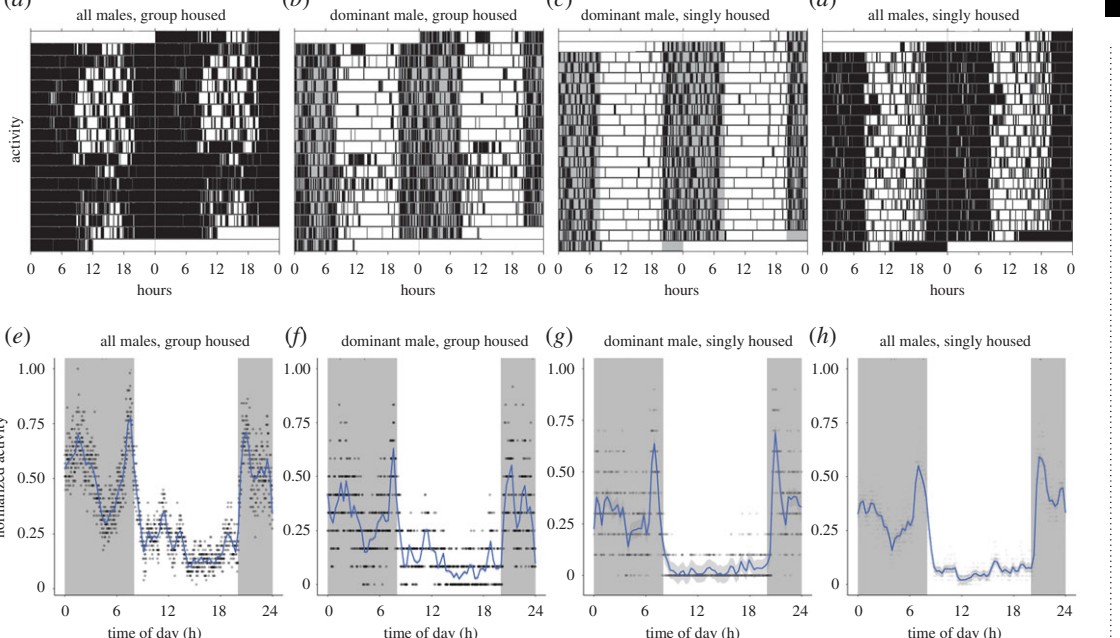

**Figure 1.** The double-plotted actograms of the total activity of six group-housed male mice (*a*), of the activity of the highest ranking of those six mice while group housed (*b*), the activity of that same mouse while singly housed (*c*) and the total activity of six male mice while singly housed (*d*). White background denotes light; grey background denotes darkness and black bars denote activity (as measured by passing through the gate between the two compartments. Note that for each animal, daytime activity is reduced under single housing conditions when compared with group housing, as is shown in figure 2*b* (see electronic supplementary material, figure S3 for the individual data underlying figure 2*a,b*). Activity is also displayed as the total relative activity over the course of the 24 h period, relative to the maximum activity. Activity has been pooled in 1 min bins over the course of the group housing phase for all individuals (*e*), for the highest ranking animal (*f*), in the single housing phase for the highest ranking animal (*g*) and in the single housing phase for all individual mice (*h*).

None of the eight dominance hierarchies had unknown or tied ranks. Linearity was ascertained using triangle transitivity [54]. Using 1000 randomizations, all the hierarchies were linear, differing significantly from randomness (for the males: $n = 4 \times 6$, $p = 0.042$, $p = 0.021$, $p = 0.045$, $p = 0.01$; for the females: $n = 4 \times 6$, $p = 0.029$, $p = 0.028$, $p = 0.024$, $p = 0.038$). The hierarchy steepness based on David's scores [54] ranged from 0.53 to 0.76 for the males, and from 0.44 to 0.60 for the females.

No interactions were observed where animals prevented other animals from accessing food, water or the nest.

## 3.2. Activity patterns: male mice

All of the male mice—and thus the group of male mice as a whole—had a predominantly nocturnal activity pattern, with an increase in activity just before the lights went on (i.e. just before the start of daytime); an example double-plotted actogram of six male mice monitored during the group housing period is shown in figure 1*a*. In general, the activity of the dominant mouse under group housing conditions looks similar to the group's activity pattern, as judged by sight from the actogram (figure 1*b*). Interestingly, when moved to single housing, daytime activity of the mice decreased markedly compared with when the mice were group housed (figure 1*c*, compare with figure 1*b* for the highest ranking mouse, figure 1*d*, compare with figure 1*a* for the whole group). All mice show this decrease (Wilcoxon signed-rank test, $n = 4 \times 6$, $V = 236$, $p < 0.0001$), but figure 2 suggests the effect is significantly stronger in the two highest ranking mice than in the other four (Mann–Whitney $U$-test $W = 101$, $n = 4 \times 6$, $p = 0.02297$), though there is no indication why the top ranking mice should react more strongly.

When switching from group to single housing, activity also decreased during the dark period (Wilcoxon signed-rank test, $n = 4 \times 6$, $V = 233$, $p = 0.0001769$). The ratio of daytime activity over total (daytime plus night-time) activity does, however, decrease when switching the males from group housing to single housing (Wilcoxon signed-rank test, $V = 246$, $p < 0.0001$).

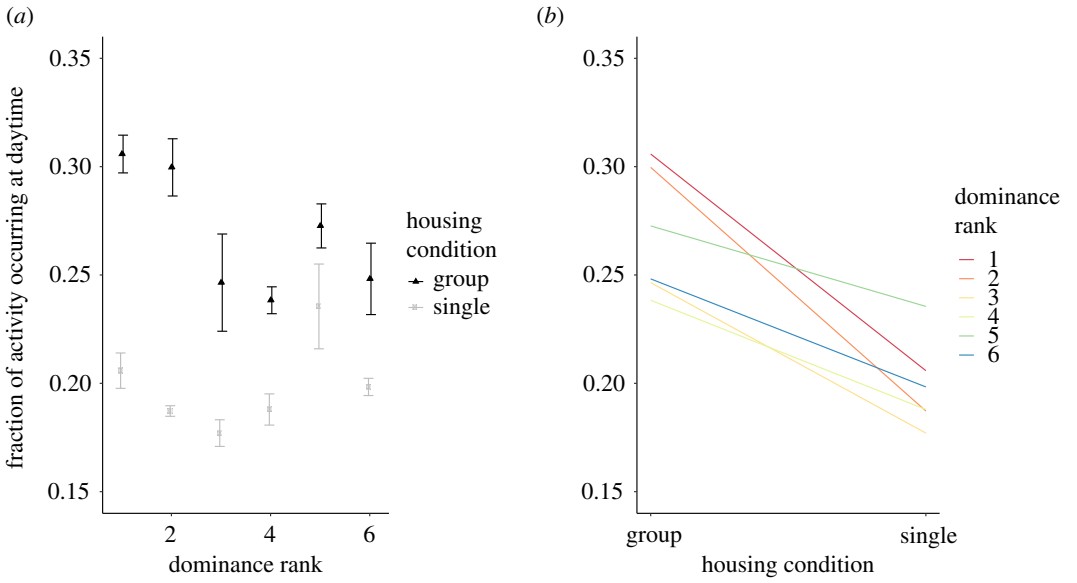

**Figure 2.** (a) Activity ratio of male mice, i.e. the fraction of daily activity occurring during the light phase (e.g. an activity ratio of 0.3 means that 30% of all activity in a 24 h period occurred during the light phase) plotted against dominance rank for six male mice group housed (triangles) and singly housed (squares). Note that for each animal, the activity ratio drops when they are changed from group housing to single housing, meaning a smaller fraction of their activity occurs during the light phase. The change is largest for the two highest ranking males. Error bars designate standard errors. (b) A derivative of figure 2a to emphasize the trend: the changes in activity ratio for each dominance rank, averaged over all four experiments with male mice. Dots on the left-hand side designate activity ratio under group housing conditions; dots on the right-hand side designate activity ratio under single housing conditions. Coloured lines connect the two dots (group housing and single housing) for each dominance rank. The colour of the line designates dominance rank, ranging from red (highest ranking) via orange, yellow, green and blue to purple (least dominant). The same error bars as displayed in figure 2a are applicable to figure 2b, but are left out for visualization purposes. The data for each replication can be found in electronic supplementary material, figure S2.

An analysis of normalized activity revealed a trough in activity during the day time and a peak in activity just after the start of the dark period, with a second peak in activity just before the start of the light period (figure 1e for group housing and figure 1h for individual housing). The individual mice had a similar pattern of activity, as shown for the dominant mouse in this group (figure 1f for group housing, and figure 1g for individual housing).

Next, we examined the ratio of activity during the light phase compared with the entire 24 h period (figure 2a). When male mice were group housed, dominance rank correlated with activity during the light phase, i.e. more dominant mice were more active than subordinate mice (Kendall's rank correlation $n = 4 \times 6$, $\tau_b = -0.3264$, $z = -2.1177$, $p = 0.03420$). After the mice were transferred to single housing, the correlation between dominance and daytime activity was no longer significant (Kendall's rank correlation $n = 4 \times 6$, $\tau_b = 0.02351$, $z = 0.1437$, $p = 0.8857$). The analysis was performed on the four replications with male mice, but in each of the four replications, all six males showed the same pattern in reduction of activity and daytime activity ratio when being switched from group housing to single housing.

## 3.3. Activity patterns: female mice

The female mice had an activity pattern that was similar to the male mice (figure 3a–d), with the highest activity occurring during the dark period. In addition, when group housed, the activity peaks among the female mice occurred at the same times as among the male mice (compare figure 3e with figure 1e and figure 3f with figure 1f). When the female mice were transferred to single housing, the qualitative pattern was largely retained (compare figure 3g with f and figure 3h with e). For females, the activity during the light phase was reduced after the move to individual housing (Wilcoxon signed-rank test $n = 4 \times 6$, $W = 1158$, $p = 0.1096$), whereas activity during the dark phase increased significantly (Wilcoxon signed-rank test, $n = 4 \times 6$, $W = 1234$, $p = 0.0254$). Nevertheless, the total activity among the female mice was similar

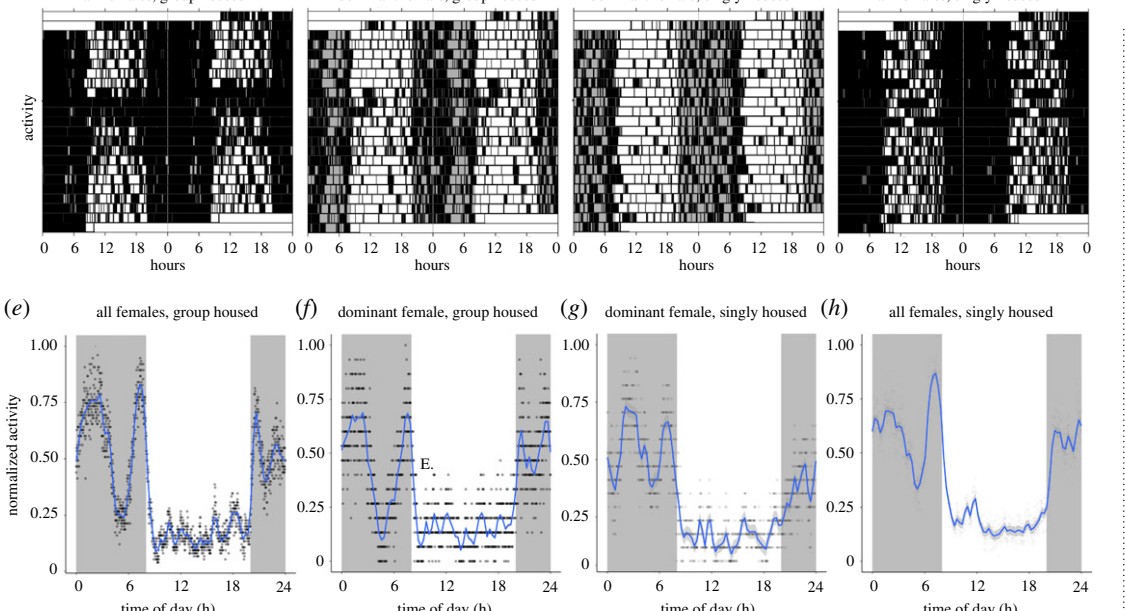

**Figure 3.** The double-plotted actograms of the total activity of six group-housed female mice (*a*), of the activity of the highest ranking of those six mice while group housed (*b*), the activity of that same mouse while singly housed (*c*) and the total activity of six female mice while singly housed (*d*). White background denotes light; grey background denotes darkness and black bars denote activity (as measured by passing through the gate between the two compartments. Note that for each animal, daytime activity is reduced under single housing conditions when compared with group housing, as is shown in figure 4*b* (see electronic supplementary material, figure S4 for the individual data underlying figure 4*a,b*). Activity is also displayed as the total relative activity over the course of the 24 h period, relative to the maximum activity. Activity has been pooled in 1 min bins over the course of the group housing phase for all individuals (*e*), for the highest ranking animal (*f*), in the single housing phase for the highest ranking animal (*g*) and in the single housing phase for all individual mice (*h*).

(i.e. did not differ significantly) between group housing and individual housing conditions (Wilcoxon signed-rank test, $n = 4 \times 6$, $W = 1194$, $p = 0.05731$).

Unlike for the male mice, we found no significant change in daytime activity when the female mice were transferred to individual housing (figure 4*a*; Wilcoxon signed-rank test, $n = 4 \times 6$, $W = 798$, $p = 0.1616$). Furthermore, and also unlike the male mice, we found no clear correlation between daytime activity and social dominance rank among the female mice while group housed (Kendall's rank correlation $n = 4 \times 6$, $\tau_b = -0.03391$, $z = -0.2148$, $p = 0.8299$). When the female mice were transferred to single housing, this remained the case (figure 4*b*; Kendall's rank correlation $n = 4 \times 6$, $\tau_b = 0.09692$, $z = 0.5837$, $p = 0.5594$).

## 3.4. Detrended fluctuation analysis

Among the male mice, the DFA $\alpha$-value was significantly closer to 1 (generally considered better) under group housing than under single housing (0.90 versus 0.87, respectively; Wilcoxon signed-rank test, $n = 4 \times 6$, $V = 241$, $p < 0.0001$). While this difference is small, $\alpha$ can vary only between 0.5 and 1.5, and small alterations are considered non-trivial [47].

By contrast, although $\alpha$ was similar in male and female mice under group housing (0.91 for females), $\alpha$ did not change after the female mice were transferred to single housing (0.91; Wilcoxon signed-rank test, $n = 4 \times 6$, $W = 64$, $p = 0.8424$).

For the group-housed male mice, $\alpha$ correlated with dominance rank (Kendall rank correlation $n = 4 \times 6$, $\tau_b = -0.3575$, $z = -2.3194$, $p = 0.02037$), indicating that higher ranking males have a DFA value closer to 1. When singly housed, this correlation disappears (Kendall rank correlation $n = 4 \times 6$, $\tau_b = -0.06961$, $z = -0.4302$, $p = 0.6670$). For the females, there is no correlation either when group housed (Kendall rank correlation $n = 4 \times 6$, $\tau_b = -0.08477$, $z = -0.5371$, $p = 0.5912$) or when singly housed (Kendall rank correlation $n = 4 \times 6$, $\tau_b = 0.04591$, $z = 0.2765$, $p = 0.7822$).

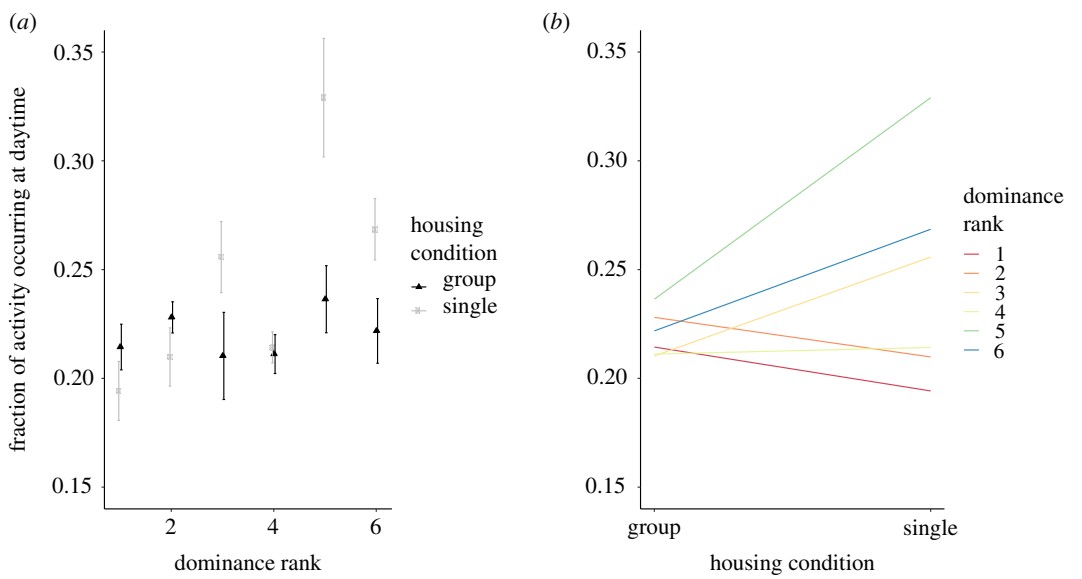

**Figure 4.** (a) Activity ratio of female mice, i.e. the fraction of daily activity occurring during the light phase (e.g. an activity ratio of 0.3 means that 30% of all activity in a 24 h period occurred during the light phase) plotted against dominance rank for six female mice group housed (triangles) and singly housed (squares). Note that for females—as opposed to males—there is no significant change in the activity when animals are changed from group housing to single housing. Error bars designate standard errors. (b) A derivative of figure 4a to emphasize the trend: the changes in activity ratio for each dominance rank, averaged over all four experiments with female mice. Dots on the left-hand side designate activity ratio under group housing conditions; dots on the right-hand side designate activity ratio under single housing conditions. Coloured lines connect the two dots (group housing and single housing) for each dominance rank. The colour of the line designates dominance rank, ranging from red (highest ranking) via orange, yellow, green and blue to purple (least dominant). The same error bars as displayed in figure 4a are applicable to figure 4b, but are left out for visualization purposes. The data for each replication can be found in electronic supplementary material, figure S4.

## 3.5. Plasma corticosterone

Next, we measured plasma corticosterone levels in both the male mice (figure 5a) and the female mice (figure 5b) under group housing and single housing at two separate times during the daily cycle 1 h after lights-on (ZT1) and 1 h before lights-off (ZT11). Our analysis revealed that corticosterone levels were higher at ZT11 than at ZT1 in both the male and female groups, regardless of their housing condition (males: Wilcoxon signed-rank test, $n = 3 \times 6$, $V = 9$, $p < 0.0001$; females: $n = 4 \times 6$, $V = 5$, $p < 0.0001$; this finding is consistent with an increase in corticosterone levels rise prior to the start of the active period [55]. Of note, the corticosterone levels at ZT1 were significantly lower in the male mice after they were transferred to single housing (Wilcoxon signed-rank test, $n = 3 \times 6$, $V = 139$, $p = 0.001678$), but were unchanged in the female mice after they were transferred to single housing (Wilcoxon signed-rank test, $n = 4 \times 6$, $W = 128$, $p = 0.5474$). Lastly, we found a significant correlation between social dominance rank and plasma corticosterone levels among group-housed males at ZT1 (Kendall rank correlation $n = 3 \times 6$, $\tau_b = 0.4070$, $z = 0.4330$, $p = 0.6650$) but not at ZT11 and not when singly housed. For females, no significant correlation between corticosterone levels and social dominance rank was found under any of these circumstances. The difference between the corticosterone levels at ZT11 and ZT1 did not significantly change for either males or females when switching from group housing to single housing, nor did it correlate with rank for either sex in either housing condition.

## 3.6. Weight

We also measured relative body weight (expressed relative to the start of the experiment) for all of the mice. We found that in absolute weight, all male mice gained weight while group housed, which is to be expected as C57Bl/6 J mice continue to gain weight at least till six months of age, and often throughout their life (Kendall's rank correlation $n = 4 \times 6$, $\tau_b = 0.5100$, $z = 8.0526$, $p < 0.0001$) [34]. Moreover, the more dominant mice gained significantly more relative weight compared with the

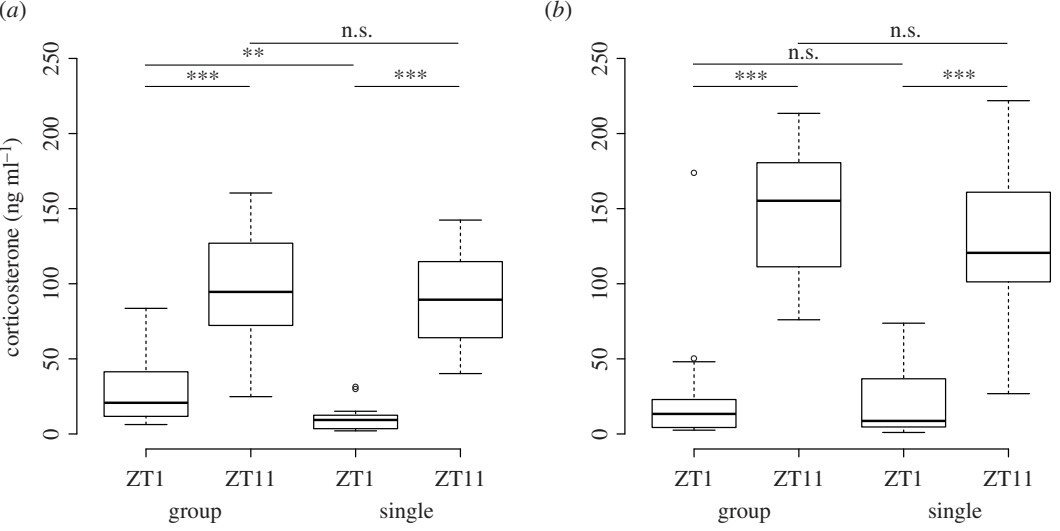

**Figure 5.** Corticosterone plasma levels for male (*a*) and female (*b*) mice during group and single housing at zeitgeber time (ZT)1 and ZT11. The corticosterone plasma levels are significantly higher at ZT11 compared with ZT1 for both male and female mice, group and single housed. For male mice, not for female mice, the corticosterone concentration at ZT1 is lower during single housing compared with group housing. Error bars designate standard errors.

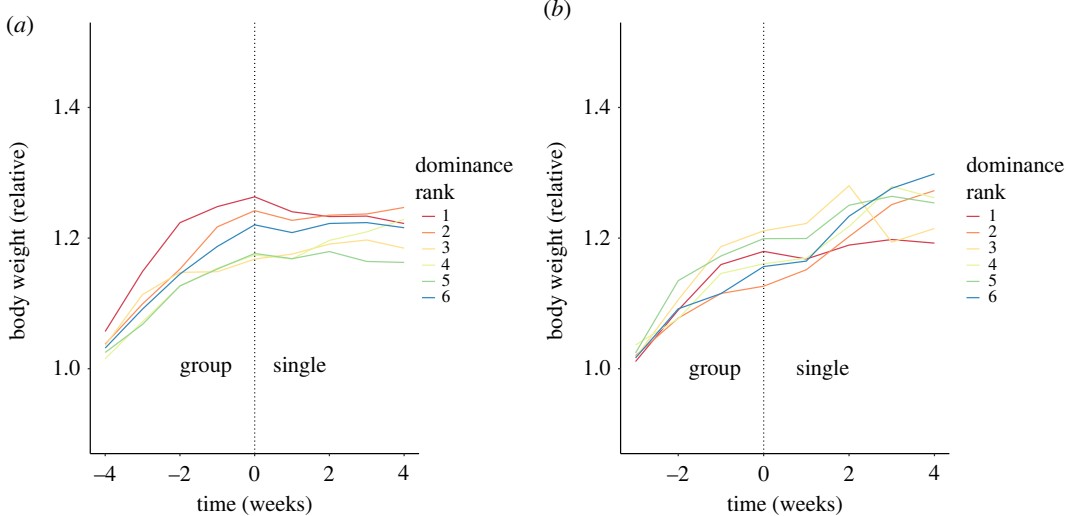

**Figure 6.** The relative development in body weight for male (*a*) and female (*b*) mice during group and single housing. The transfer from group housing to single housing is demarcated by a vertical dotted line. For male mice, not for female mice, the increase in weight is halted at the time of transfer. The colour of the line designates dominance rank, ranging from red (highest ranking) via orange, yellow, green and blue to purple (least dominant). The highest ranking male mice during group housing has the fastest increase in body weight compared with the other ranks.

subordinate mice (figure 6*a*; Kendall's rank correlation $n = 4 \times 6$, $\tau_b = -0.1751$, $z = -3.5529$, $p = 0.0003810$), consistent with previous reports [18]. After they were transferred to single housing, each male mouse had a small yet significant drop in weight (Wilcoxon signed-rank test, $n = 4 \times 6$, $V = 36$, $p = 0.007813$), followed by a plateau in weight that lasted throughout the remainder of the study. Since the group housing varied from four to six weeks for the male mice, yet the drop in weight always occurred in the week after the change to single housing, it can be concluded that this drop in weight is caused by the transfer from group to single housing rather than by an age effect. The correlation between dominance rank and relative body weight gain was no longer significant one week after the mice were transferred to single housing (Kendall's rank correlation $\tau_b = -0.002248$, $z = -0.03793$, $p = 0.9697$).

Under group housing, the female mice gained absolute weight to the same extent as the male mice (Mann–Whitney $U$-test, $n = 4 \times 6$, $W = 5249$, $p = 0.06214$ versus the weight gain in the male mice). In

contrast with the male mice, however, the female mice continued to gain absolute weight after they were transferred to single housing (Wilcoxon signed-rank test, $n = 4 \times 6$, $V = 193$, $p = 0.09740$). There was no correlation between dominance rank and relative weight gain for female mice, nor while group housed (figure 6b; Kendall's rank correlation, $n = 4 \times 6$, $\tau_b = -0.01688$, $z = -0.2118$, $p = 0.8323$), nor while singly housed (Kendall's rank correlation, $n = 4 \times 6$, $\tau_b = 0.03927$, $z = 0.51143$, $p = 0.6091$), unlike the situation for males.

## 4. Discussion

Here, we examined the activity of group-housed mice in order to investigate the effect that their position in the social dominance hierarchy has on activity. Specifically, we tested the hypothesis that subordinate mice increase their daytime activity due to food restrictions imposed by their dominant nest-mates. In contrast with this hypothesis, we found a significant correlation between social dominance rank in males and the relative amount of activity during the day time, with dominant males being more active during the day time compared with subordinate males; we found no such correlation among female mice. After transferring to individual housing, the amount of daytime activity decreased in all male mice, resulting in all male mice exhibiting similar levels of daytime activity, irrespective of their previous social dominance ranking established while group housed. It does fit in with findings in the literature that dominant male mice living in hierarchies have consistently higher levels of activity and will feed and drink all day [56]. Given that we found no evidence of increased stress, we speculate that the relatively higher daytime activity among the dominant group-housed mice may represent a coping strategy intended to reduce stress when the population density is high, and is, therefore, part of normal social biological behaviour.

Our study indicates that social hierarchy in mice can influence their pattern of circadian activity. Interestingly, however, these effects of social behaviour on the circadian system differed markedly between female and male mice, even though both sexes established a similar dominance hierarchy when group housed. The female mice lacked the effects of dominance rank on their 24 h activity patterns that we observed in male mice, as we found no correlation between dominance rank among females and their activity patterns.

The activity pattern of cohabitating rodents has been a topic of investigation in the past (1950–1970) [31,57–60], as has—more recently—their sleep pattern and quality (2019) [10]. While it was noted that temporal segregation takes place, the inverse relation between temporal niche shifting and dominance rank was not predicted in these papers. In a year-round study in the wild, we have previously observed that daytime activity increases when population density increases [27]. We intuitively expected that subordinate animals would avoid the dominant animals and, therefore, shift to relatively more daytime activity. Apparently, this expectation was incorrect, and instead particularly the dominant males were more active during the day under group housing than subordinate animals. As soon as the male animals were housed individually, their daytime activity dropped, and the circadian rhythms of dominant and subordinate mice were indistinguishable.

For female animals, the group housing did not affect their temporal profile. A difference between males and females was the increased number of agonistic encounters in female mice, which is likely to reflect a higher sociability state of the females. It was surprising, therefore, that the circadian patterns of female mice were not different under group housing versus individual housing and were also not dependent on social dominance rank.

Previous studies involving social hierarchies in mice reported the presence of a single alpha mouse, with subordinate nest-mates [15,16,20,21] bearing bite marks due to the aggressive behaviour of the group. In our study, we did not observe this level of aggressive behaviour, probably because we used C57BL/6 J mice, a strain known to be relatively non-aggressive [15,16,61–64]. Furthermore, the groups of mice in our study had never been housed individually prior to our experiments; by contrast, most studies that focus on social dominance house the mice together following a period of isolation [65–67]. Although Fairless et al. [65] found that sociability among group-housed mice is similar between litter-mates and non-litter-mates, their prior isolation may have contributed to the aggression reported in previous studies [65,67].

Importantly, the type of data collected in our study allowed us to identify the stratification of ranks in a manner that past papers could not always do [15]. Our methods are similar to those of Williamson et al. [68], although they used Glicko ratings rather than the Elo rating used in our study. Either rating system allows one to attribute a quantitative estimation of the degree of dominance for all of the mice in the

group, rather than identifying only one purely dominant mouse. Thus, although one had the highest dominance rating and won most agonistic interactions with all of the other animals, a complete dominance hierarchy was identified, allowing us to test for correlations between dominance rank and other variables. Thus, our findings suggest that the social dominance hierarchy among both male and female groups is more nuanced than the presence of one dominant mouse and a group of subordinate mice as well as that female mice form hierarchies just as male mice do. These findings match with those from Williamson *et al.* and several other papers reviewed by them [20], most importantly the paper by Been *et al.* [69]. Notably, this study confirms the finding that female mice also have dominance hierarchies [20], [21, ch. 4], [30,70].

Nevertheless, why dominant male mice are relatively more active during the day when group housed than when housed individually remains an open question. Van der Vinne *et al.* [71] reported that daytime activity increases in mice when energetically challenged, for example, by cold or hunger. The authors proposed the circadian thermoenergetics hypothesis as a possible explanation for this phenomenon. According to this hypothesis, cold and hunger induce a negative energy balance in mice, which can be corrected by increasing their activity during the day, when ambient temperature is higher. However, in our study, the temperature in the cages was held relatively constant and food was available ad libitum, making this hypothesis an unlikely explanation for our observations.

Another open question is whether 'stress' itself may have caused the behavioural activity patterns, particularly among the dominant male mice. For example, Palanza *et al.* [72] found that social stress has distinct causes—as well as distinct effects—in male and female mice, and these differences may have affected the two sexes in our study differently. To address this question, we measured plasma corticosterone levels as an indicator of perceived stress and attempted to find a correlation with dominance status; however, our analysis revealed no clear correlation between social rank and corticosterone levels among either the males or females. Previous findings described in the literature are not consistent: some studies do not find a link between plasma corticosterone levels and the position in the dominance hierarchy [19], while other studies indicate that there can be a link between dominance rank and plasma corticosterone levels in female mice [20], [21, ch. 1, 4 and 9], though the referenced study made use of CD-1 mice that are more aggressive than the C57BL/6 J mice used in this study [15,61–64]. From our study, we conclude that the different group members had no obvious differences in corticosterone levels that would explain the behavioural activity patterns.

Another possible explanation is that dominant male mice are more active during the day—compared with subordinate mice—to maintain their dominant position. Daytime activity may be required in order to preserve their position within the group, leading to a continuous state of alertness in dominant animals, not only during the night but also during the day. This would offer an explanation for our counterintuitive finding that not the subordinate, but rather the dominant male mice are most active in the daytime niche.

Daytime activity in nocturnal animals is strongly associated with ageing, disease, overweight and with a sedentary lifestyle [73–76]. Also, in humans, activity during the 'wrong' part of the cycle is characteristic for a decrement in the functioning of the circadian system and is related to a broad range of diseases [77]. Although daytime activity can be detrimental in nocturnal animals, the ensemble of the measures of corticosterone, growth rate and DFA score indicated no obvious health differences between our dominant and subordinate mice. Thus, we conclude that the decrease in circadian rhythmicity that emerges under group housing reflects biological adaptive behaviour, that arises at the 'group' level, and is in no way pathological. Food for thought is the observation that from single cell, to network, to multi-organismal level, it is predominantly a decrease in rhythm strength that is achieved.

Ethics. All animal experiments were approved by the institutional ethics committee on animal care and experimentation at Leiden University Medical Centre (LUMC), Leiden, The Netherlands.

Data accessibility. Data available from the Dryad Digital Repository: https://doi.org/10.5061/dryad.s1rn8pk65 [38].

Authors' contributions. Y.R., C.P.C. and J.H.M. designed the experiments. Y.R., C.P.C. and M.M.H.T. performed the experiments. Y.R. and C.P.C. analysed the data. Y.R., J.H.M. and C.P.C. wrote the paper.

Competing interests. The authors declare that no conflict of interest exists.

Funding. This research was funded by the Dutch Diabetes Research Foundation (grant 2013.81.1663 to C.P.C.) and by the Dutch Foundation for Scientific Research (NWO), grant no. 023.001.087 to Y.R. and J.H.M.

Acknowledgements. The authors thank Jan A.M. Janse, Cedric R.L. de Wijs and Marfa L. Blanter for providing technical assistance and Oliver Schülke and James Curley for providing valuable feedback.

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
