## [Peer Review File · Royal Society Open Science]

Review History

RSOS-192237.R0 (Original submission)

Review form: Reviewer 1

Is the manuscript scientifically sound in its present form?

Yes

Are the interpretations and conclusions justified by the results?

Yes

Is the language acceptable?

Yes

Do you have any ethical concerns with this paper?

No

Have you any concerns about statistical analyses in this paper?

No

Recommendation?

Accept with minor revision (please list in comments)

Comments to the Author(s)

This is a very interesting study that investigates the relationship between dominance rank and activity levels across the light cycle in male and female mice. The paper is noteworthy as the authors demonstrate that dominant males show increased daytime activity compared to subordinate mice, but only when group housed. The authors also show that female mice form hierarchies with individual ranks, but that the patterns of activity are not different by rank. The authors use RFID and video methods to gather high quality data to support their findings. I enjoyed reading the paper and believe it to be technically sound. I think it's a welcome addition to the literature. My major comments are however to do with some literature that the authors did not cite that actually supports their findings and conclusions. I feel that I (James Curley, UT Austin) should identify myself as the reviewer, as the majority of the papers I believe the authors should cite are from my own laboratory. I hope the authors and editor do not mind me doing this - I really am not in the habit of asking authors to cite my own work - but I feel that these papers share important findings that should be included.

1. In the introduction and discussion, the authors note papers that investigated how social group composition affects activity patterns, but they note that the relationship between dominance rank and temporal niche shifting was not investigated. To this end, the authors have not cited findings from two recent studies that need to be included. In one case, results from this study support their own findings.

i) Lee et al 2018, Foraging dynamics are associated with social status and context in mouse social hierarchies, PeerJ. <https://www.ncbi.nlm.nih.gov/pubmed/30258716>. This paper demonstrates that dominant male mice living in hierarchies show consistently higher levels of activity and feeding/drinking throughout the 24hour period, including during the light period, compared to subordinate mice. These findings are congruent with the current study's findings that dominant mice increase their daytime activity when group housed. The proposed reason for the increased daytime activity by dominant mice in the Lee et al paper is that this is due to increased behavioral (patrolling and territorial defense) and energetic demands of being dominant - as argued on line 12 of page 12.

ii) Karamihalev et al 2019, Social context and dominance status contribute to sleep patterns and quality in groups of freely-moving mice, Sci Rep. <https://www.nature.com/articles/s41598-019-51375-7> This paper shows that dominance status influences sleep quality. Both of these papers speak importantly to the issues and questions raised in the current paper and should be included.

2. The findings that females form social hierarchies. This is a very important finding as so few studies have investigated whether female mice form hierarchies. This confirms findings in Williamson et al 2019 Sci Reports <https://www.nature.com/articles/s41598-019-43747-w>, that females can indeed form hierarchies. In that paper we also review the few other studies that have investigated that question. The current study has excellent quality data that provide strong support for the hypothesis that female mice do indeed form hierarchies.

3. In the discussion (page 11 lines 13-38), the authors describe the type of hierarchies that mice form. They discuss how they are usually described as having one dominant and several subordinate animals. They state that their ELO rating method enables them to identify individual ranks. I would argue that the reason why previous studies reported one dominant animal and several subordinate is largely due to the type of data that was collected meaning

those authors were unable to identify the stratification of ranks. In our own work (e.g. see Williamson et al 2016 "Temporal dynamics of social hierarchy formation and maintenance in male mice" <https://www.sciencedirect.com/science/article/pii/S0003347216000798> we applied a Glicko method (essentially an improved version of ELO) that we used to identify individual ranks in mice. It should be mentioned that current studies of social hierarchies in mice are revealing that mice do form hierarchies with individual ranks being identifiable. Further, the use of ELO is not actually that strong a method for identifying clearly different rank orders. To do that, you should technically use I&SI ranking (although this does correlate well with ELO ranking). We have used I&SI rankings in a series of papers to show that mice do indeed form hierarchies with individual ranks. Although in the current study, I think that ELO is sufficient. All this is to say, that I believe the authors excellent high quality data is confirming these findings, which is very welcome, and suggests that the older literature were assuming a pattern of dominance relationships without being able to detect the more detailed gradations in ranks.

Review form: Reviewer 2

Is the manuscript scientifically sound in its present form?

Yes

Are the interpretations and conclusions justified by the results?

Yes

Is the language acceptable?

Yes

Do you have any ethical concerns with this paper?

No

Have you any concerns about statistical analyses in this paper?

No

Recommendation?

Accept with minor revision (please list in comments)

Comments to the Author(s)

Reviewer comments to the manuscript 'Group housing and social dominance hierarchy affect circadian activity patterns in mice' by Yuri Robbers, Mayke M. H. Tersteeg, Johanna H. Meijer, and Claudia P. Coomans

Robbers et al. report the impact of group housing and dominance on the daily activity of C57BL/6 mice. It is highly interesting how individual species adapt their activity pattern to their social environment, and relatively little is known about this, both in the lab and in the field. I therefore like this study; the approach is very basic and the outcome can and should be published.

Generally, I do not have many comments on the manuscript. The paper is well written and to the point. The only, minor comment - or actually question - I have, is on the analysis - I wonder why the authors did not choose to use non-parametric models with individuals nested in their social groups? I however realize that the approach as chosen now should be more conservative, and probably does not differ much for for example the near significant overall difference in activity of group- and single housed females.

I have no further comments; please check symbol indications in figure descriptions

Decision letter (RSOS-192237.R0)

Dear Dr Meijer:

Manuscript ID RSOS-192237 entitled "Group housing and social dominance hierarchy affect circadian activity patterns in mice" which you submitted to Royal Society Open Science, has been reviewed. The comments from reviewers are included at the bottom of this letter.

In view of the criticisms of the reviewers, the manuscript has been rejected in its current form. However, a new manuscript may be submitted which takes into consideration these comments.

Please note that resubmitting your manuscript does not guarantee eventual acceptance, and that your resubmission will be subject to peer review before a decision is made.

Your resubmitted manuscript should be submitted by 15-Nov-2020. If you are unable to submit by this date please contact the Editorial Office.

on behalf of Dr Oliver Schülke (Associate Editor) and Kevin Padian (Subject Editor)
openscience@royalsociety.org

Subject Editor Comments to Author (Professor Kevin Padian):
Comments to the Author:

Thanks for your submission. Both reviewers liked it, although one had suggestions on other work that should be cited. Our AE devoted considerable time to reviewing it and came to a less sanguine conclusion, as you will see from that review. If you would like to resubmit, please take

your time to address all points carefully, and tell us what in summary you have done to modify the paper for resubmission. Best wishes.

Associate Editor Comments to Author (Dr Oliver Schülke):
Comments to the Author:

Dear Dr. Meijer,

as the associate editor handling your submission I have now received comments from two referees. Since both reviews lack the necessary detail, I provide an independent review myself. Based on the comments provided below I will recommend my editor to reject the paper with the possibility of resubmission. My decision is mainly based on the failure to provide access to the data (the dryad link is dysfunctional) and problems with the description of the statistical methods employed that make it impossible to judge the paper in its current state along with shortcomings in the introduction and discussion. I am certain that you will be able to remedy the issues raised below and by the reviewers and I hope to be able to reconsider soon a substantially revised version of your work.

With my best regards,
Oliver Schülke

Introduction

The introduction after 90 words lands very quickly at lab mice without providing a broader introduction either to the behavioral ecology of the mouse, to the evolution of dominance hierarchies, to temporal niche segregation, or to density effects on grouping patterns as observed for example in wild cavies (the wild form of guinea pigs) and many other animals. This is especially relevant for this journal since it addresses a very broad readership.

P4, Line 29: Please provide a reference that established the term "Fujimoto effect".

P4, Line 33: Please explain what the crucial difference is between previous analyses and yours. What is circadian timescale? And consider here the work by one of your reviewers.

P4, Line 39: Please explain what the costs and benefits of activity synchronization would be to provide some background and how these costs and benefits would change with density or grouping pattern.

Most of the material presented on Lines 45ff needs to be moved to the methods and results section. It should be replaced by clear predictions derived from previous work and theory and a justification for the use of each response variable. The reader needs to be prepared for what is to come. Corticosterone measurements are mentioned in the last sentence but their use is not justified. Why is activity measured with four different activity indices derived from the RFID data – what is the added benefit and how do they relate to the research question and the Fujimoto effect? Why are they relevant here and what are they implied to measure? Why measure weight and what do you predict?

Methods

The methods section lacks detail.

It is unclear how individual body mass was assessed.

P5, Line 14: Please let the reader know whether 8 week old mice are adult.

P5, Animals: The explanation of the design seems complicated because the replicates are labelled as experiments. There was only one type of treatment, i.e. to move group housed mice into individual cages and therefore it seems more straightforward to construe this as one experiment. The experiment consisted of observations of eight groups of six same-sex mice while house together and again after being moved to individual cages with half of the groups comprising only males and half only females.

P5, Line 35ff: The use of four measures of activity needs to be justified in the introduction.

P5, Line 44ff: It is unclear whether the data point that entered the Elo rating was a full conflict from the start to its end culminating in one mouse showing clear submission or the behavioral element. Listing all aggressive and submissive behaviors and then saying "Each of these behaviors was considered an interaction" suggest the latter which may inflate sample size for hierarchy construction and therefore inter-individual differences in winning success.

P5, Line 60: Consider rephrasing. It was not the mice that were analyzed but their behavior.

P6, Line 18: Please provide a reference that validates the EIA for the use with blood from mice.

P6, Line 25: Please explain how you determined when the hierarchy was established and be very clear about whether you refer to the mice needing time or your tool needing time which is conceptually very different. If you refer to the burn-in period, there has been considerable progress in the development of the method in the past three years (see below).

Please provide more detail on your Elo-rating including starting values and k values used as well as how you determined the burn in phase. Newton-Fisher et al 2017 (<https://doi.org/10.1007/s10764-017-9952-2>) have improved the Elo-rating method suggested by Albers & De Vries, Sanchez-Tojar et al. 2017 provide a guide for assessing the reliability of hierarchies constructed with different methods (DOI: 10.1111/1365-2656.12776), Goffe et al 2018 (DOI: 10.1111/2041-210X.13072) provide another objective method for determining the burn-in period, Vilette et al (<https://www.biorxiv.org/content/10.1101/692384v3>) for a guide for choosing the right ranking method for a particular research question.

Please provide the number of conflicts that entered the hierarchy analysis for each group as a quality control for the hierarchy.

The description of the statistical methods used is too scarce to provide a clear picture.

How did you deal with the dependency of data collected in the same group? It remains unclear whether data for one group (or one rank position) have been averaged before entering a test or whether a data point per individual was used in all analyses presented. Had samples sizes been reported with the test results, such confusion could have been avoided. Both approaches have their problems: if data were averaged per dominance rank before running correlations it is possible that a relationship was present in one or two groups but not the others. If the correlation was run on a sample of 4x6 individuals with 4 alphas and 4 betas and so forth, the biological dependencies introduced by housing the individuals in groups are ignored and samples size is unduly inflated. More description of the data are needed to judge whether hidden diversity between groups may have hampered the results based on averaged values.

Which of your Wilcoxon rank-sum tests was a paired test and which was a test for independent samples and thus equivalent to a Mann-Whitney U-test. From my reading of the manuscript it seems that both types of tests have been used. Is it true that the test is referred to as a signed rank test if paired and as rank sum test if not?

It is unclear which software has been used to run the tests and whether the adjustments required for both types of tests when run on small samples sizes have been used (Mundry & Fischer 1998 *Anim Behav* 56, 256-259).

Results

Page 7, Line 10: It is unclear why and how a dominant individual or two dominant individuals have been determined for every group. Elo-rating provides a continuous measure of dominance without an objective threshold for determining who is dominant and who is subordinate.

If some analyses are run across ranks, why run comparisons between the alpha and the rest? Is there a good reason to expect dominance to not be linear but classifying? If so, this needs to be introduced early on and a method be devised for such classification. The description here makes it sound like this was done by behavioral type (these mice were identified by their aggressive nature) and not by dominance rank or Elo-score. This is confusing.

P7, Line 22: It is unclear whether the interaction rate has been calculated per individual per hour or per group per hour.

P7, Line 27: It is unclear, how it was determined when a stable dominance hierarchy was established. See above for the conceptual difference between the animals establishing a stable hierarchy and our methods finding it.

P7, Line 42: How was it established that the dominant mouse (is it nor true that there were sometimes 2 of them) had a similar activity pattern to the other group members.

P7, Line 43: The sentence starting on line 43 has the same content as the one starting on line 48 and therefore is redundant and can be omitted.

P7, Line 49: The high value of W and the very low p value suggest that the test has been run on all 4x6 male mice in matched design which is problematic because of the non-independence of data point coming from the same group. Perhaps one could provide data per group in the supplementary material to double check whether the pattern is replicated in every group.

P7, Line 51: If the signed rank test is indeed the matched version of the Wilcoxon test, then it has not been applied correctly here for the comparison between dominants and subordinates which are two classes of independent data points. Why are the two top ranking males chosen here instead of either one or two in each group as suggested above was the pattern that emerged from the data. It seems that 4x2 dominants have been compared to 4x4 subordinates.

P8, Line 11: The text describes a difference where the test was run as a correlation. This is problematic here because other tests have indeed compared dominants and subordinates.

P8, Line 14: The correlation was run across all males from four groups. This is problematic because the ordinal rank variable contains a lot of ties then which needs correcting of Spearman's test and may even forbid it given that there are so many tied ranks. If it was not the ranks that entered the test but the Elo-Scores, they would need to be standardized before they can be used because Elo-Scores cannot be compared between groups. They are always relative to the other animals in the game.

Some of the confusion about what was tested stems from the fact that figure 2 and 4 present data point per rank position only which suggest that this was what entered the tests.

P8, Line 5: Please specify what the values are (means across all males or medians).

P9, Line 37: It may be revealing to run analyses on the change in CORT between ZT1 and ZT11 compared between housing conditions.

P9, Weight: Why are all mice gaining body mass? Are they still growing when 8 weeks old? What is the biological background for this measure?

Discussion

Overall, the discussion reads like a list of posthoc explanations going for one response variable to the next. If the introduction would have set up the predictions from theory, one would now why CORT or body mass were measured and how this informs the research question. Here, one learns why CORT was measured after the entire activity results have been discussed. A more integrative treatise would be desirable.

P10, Line 26: In social behavior research the term emergent property is used like in system science, as a property that is brought about by the system that is more than its parts and that is more global than individual behavioral strategies can be. In that sense, the observations described here are not emergent properties but a naturally or sexually selected strategy to deal with increased density or with living in groups. I also do not understand the contrast with a possible reaction to a stressful situation as not being part of normal social biological behavior. The stress reaction is naturally selected and an integral part of the interaction of the individual with its environment.

P10, Line 34: Characteristics of the male and female dominance hierarchy have not been described in enough detail to say that they are similar. One could test linearity, steepness, relationship consistency, proportion of tied relationships or inconsistent relationships and the like.

P10, Line 46ff: This paragraph needs to be rewritten to include the papers suggested by the reviewer.

P11, Line 3ff: Work on wild house mice suggests that male and female social organization differs markedly. Females may form close affiliative relationships with one or two other females, form a substructure to the broader grouping pattern and communally raise their young (work by Barbara Koenig from Zurich University and colleagues) whereas males establish local dominance

hierarchies and regularly roam over larger areas seeking access to as many females as possible. Moreover, there is research showing sex differences in reaction to social stress cited on lines 57ff. It is therefore not surprising that male and female reactions to co-housing are different.

P11, Line 6: Sociability is usually used to describe an affiliative tendency, seeking rewarding social contact. In this sense, it is not straightforward why increased sociability would be a consequence of higher sociability of females. If the argument is that females are in close proximity more often because of their tendency to affiliate and that therefore the *prima facie* opportunity for aggression is also increased, the argument should be supported by data on proximity or affiliation.

P11, Line 28: Here the strength of Elo-rating as a method that measured dominance on a continuous scale is highlighted as a strength. Why have so many analyses been run as comparisons between one or two dominants versus a seemingly homogenous group of subordinates? Is it intended to say that the big news is the occasional second dominant with a homogenous unstructured set of subordinates?

P11, Line 33: What does it mean for two mice to have the highest dominance rating in a group? Did they have the same rating? Given how Elo-rating works, this is extremely unlikely.

P11, Line 57: What is stress itself? Stress is the phenomenon that the organism adjusts certain systems to maintain homeostasis in reaction to a stressor. Since the introduction failed to clearly spell out the predictions derived from theory, it must remain unclear here, why differences in CORT would be expected and in which way they would go. Also, CORT is now regarded more as a mediator of solutions to adaptive problems than an “indicator of stress” (<https://doi.org/10.1016/j.yhbeh.2016.07.001>).

P12, Line 11ff: The crucial element lacking here seems to be mating competition. If males form dominance hierarchies mainly to regulate access to receptive females, dominant males may be more active as a part of their mate seeking behavior and maintenance of their dominance position.

P12, Line 29: Using activity pattern analysis (DFA) to measure the health outcome of changes in activity seems to be very indirect and perhaps even circular. Please adjust your wording here since the study has not quantified any measure of health.

References: The reference list is full of error concerning capitalization, omissions, superfluous material, and reference style.

Figures: Text in the figures is very small in figures 1 and 3. It would also be very helpful to indicate in the figure in small text boxes in or above each panel instead of just numbering them a to h.

Figure legends: Since figures 1 and 3, or 2 and 4 show the same data quality for males and females the words males and females should be placed more towards the beginning of the title. As they stand now, one has to read three lines to get to the point where one is informed about whether one looks at male or female data in figure 2 and 4.

All the line figures need a measure of spread in the data to be included. Otherwise the reader cannot appreciate how similar or different animals in the four groups were and how much neighboring ranks overlap or not. To this end figures 2a and b and figures 4 a and b could be combined into one.

Figure legend Fig 5: It is not true that “For male mice, but not for female mice, the CORT concentration is lower during single housing than compared to group housing.” This applies only to ZT1 CORT but not ZT11 CORT. Please specify.

Reviewers' Comments to Author:

Reviewer: 1

Comments to the Author(s)

This is a very interesting study that investigates the relationship between dominance rank and activity levels across the light cycle in male and female mice. The paper is noteworthy as the

authors demonstrate that dominant males show increased daytime activity compared to subordinate mice, but only when group housed. The authors also show that female mice form hierarchies with individual ranks, but that the patterns of activity are not different by rank. The authors use RFID and video methods to gather high quality data to support their findings. I enjoyed reading the paper and believe it to be technically sound. I think it's a welcome addition to the literature. My major comments are however to do with some literature that the authors did not cite that actually supports their findings and conclusions. I feel that I (James Curley, UT Austin) should identify myself as the reviewer, as the majority of the papers I believe the authors should cite are from my own laboratory. I hope the authors and editor do not mind me doing this - I really am not in the habit of asking authors to cite my own work - but I feel that these papers share important findings that should be included.

1. In the introduction and discussion, the authors note papers that investigated how social group composition affects activity patterns, but they note that the relationship between dominance rank and temporal niche shifting was not investigated. To this end, the authors have not cited findings from two recent studies that need to be included. In one case, results from this study support their own findings.

i) Lee et al 2018, Foraging dynamics are associated with social status and context in mouse social hierarchies, *PeerJ*. <https://www.ncbi.nlm.nih.gov/pubmed/30258716>. This paper demonstrates that dominant male mice living in hierarchies show consistently higher levels of activity and feeding/drinking throughout the 24hour period, including during the light period, compared to subordinate mice. These findings are congruent with the current study's findings that dominant mice increase their daytime activity when group housed. The proposed reason for the increased daytime activity by dominant mice in the Lee et al paper is that this is due to increased behavioral (patrolling and territorial defense) and energetic demands of being dominant - as argued on line 12 of page 12.

ii) Karamihalev et al 2019, Social context and dominance status contribute to sleep patterns and quality in groups of freely-moving mice, *Sci Rep*. <https://www.nature.com/articles/s41598-019-51375-7> This paper shows that dominance status influences sleep quality. Both of these papers speak importantly to the issues and questions raised in the current paper and should be included.

2. The findings that females form social hierarchies. This is a very important finding as so few studies have investigated whether female mice form hierarchies. This confirms findings in Williamson et al 2019 *Sci Reports* <https://www.nature.com/articles/s41598-019-43747-w>, that females can indeed form hierarchies. In that paper we also review the few other studies that have investigated that question. The current study has excellent quality data that provide strong support for the hypothesis that female mice do indeed form hierarchies.

3. In the discussion (page 11 lines 13-38), the authors describe the type of hierarchies that mice form. They discuss how they are usually described as having one dominant and several subordinate animals. They state that their ELO rating method enables them to identify individual ranks. I would argue that the reason why previous studies reported one dominant animal and several subordinate is largely due to the type of data that was collected meaning those authors were unable to identify the stratification of ranks. In our own work (e.g. see Williamson et al 2016 "Temporal dynamics of social hierarchy formation and maintenance in male mice" <https://www.sciencedirect.com/science/article/pii/S0003347216000798> we applied a Glicko method (essentially an improved version of ELO) that we used to identify individual ranks in mice. It should be mentioned that current studies of social hierarchies in mice are revealing that mice do form hierarchies with individual ranks being identifiable. Further, the use of ELO is not actually that strong a method for identifying clearly different rank orders. To do

that, you should technically use I&SI ranking (although this does correlate well with ELO ranking). We have used I&SI rankings in a series of papers to show that mice do indeed form hierarchies with individual ranks. Although in the current study, I think that ELO is sufficient. All this is to say, that I believe the authors excellent high quality data is confirming these findings, which is very welcome, and suggests that the older literature were assuming a pattern of dominance relationships without being able to detect the more detailed gradations in ranks.

Reviewer: 2

Comments to the Author(s)

Reviewer comments to the manuscript 'Group housing and social dominance hierarchy affect circadian activity patterns in mice' by Yuri Robbers, Mayke M. H. Tersteeg, Johanna H. Meijer, and Claudia P. Coomans

Robbers et al. report the impact of group housing and dominance on the daily activity of C57BL/6 mice. It is highly interesting how individual species adapt their activity pattern to their social environment, and relatively little is known about this, both in the lab and in the field. I therefore like this study; the approach is very basic and the outcome can and should be published.

Generally, I do not have many comments on the manuscript. The paper is well written and to the point. The only, minor comment - or actually question - I have, is on the analysis - I wonder why the authors did not choose to use non-parametric models with individuals nested in their social groups? I however realize that the approach as chosen now should be more conservative, and probably does not differ much for for example the near significant overall difference in activity of group- and single housed females.

I have no further comments; please check symbol indications in figure descriptions

Author's Response to Decision Letter for (RSOS-192237.R0)

See Appendix A.

RSOS-201985.R0

Review form: Reviewer 1

Is the manuscript scientifically sound in its present form?

Yes

Are the interpretations and conclusions justified by the results?

Yes

Is the language acceptable?

Yes

Do you have any ethical concerns with this paper?

No

Have you any concerns about statistical analyses in this paper?

No

Recommendation?

Accept with minor revision (please list in comments)

Comments to the Author(s)

I thank the authors for attending to my comments. I believe that the paper is a worthy addition to the literature.

With respect to Royal Society Open Science's data and code policy however - I was able to download and access the raw data for the paper, but I was unable to locate any supporting code. Therefore, I was unable to replicate the analyses.

Review form: Reviewer 2

Is the manuscript scientifically sound in its present form?

Yes

Are the interpretations and conclusions justified by the results?

Yes

Is the language acceptable?

Yes

Do you have any ethical concerns with this paper?

No

Have you any concerns about statistical analyses in this paper?

No

Recommendation?

Accept as is

Comments to the Author(s)

I have no further comments to the authors.

Decision letter (RSOS-201985.R0)

Dear Dr Meijer,

On behalf of the Editors, we are pleased to inform you that your Manuscript RSOS-201985 "Group housing and social dominance hierarchy affect circadian activity patterns in mice" has

been accepted for publication in Royal Society Open Science subject to minor revision in accordance with the referees' reports. Please find the referees' comments along with any feedback from the Editors below my signature.

Please submit your revised manuscript and required files (see below) no later than 7 days from today's (ie 11-Dec-2020) date. Note: the ScholarOne system will 'lock' if submission of the revision is attempted 7 or more days after the deadline. If you do not think you will be able to meet this deadline please contact the editorial office immediately.

Best regards,

on behalf of Dr Oliver Schülke (Associate Editor) and Kevin Padian (Subject Editor)
openscience@royalsociety.org

Associate Editor Comments to Author (Dr Oliver Schülke):

Dear Dr. Meijer,
I am glad to let you know that both reviewers were happy with your revisions. Both did, however, point out that you failed to put your R code on dryad as required by the journal. Once you have done so, we will accept the ms as is.
Thank you for your understanding,
Oliver Schülke
(Associate Editor RSOS)

Reviewer comments to Author:

Reviewer: 2
Comments to the Author(s)

I have no further comments to the authors.

Reviewer: 1
Comments to the Author(s)

I thank the authors for attending to my comments. I believe that the paper is a worthy addition to the literature.

With respect to Royal Society Open Science's data and code policy however - I was able to download and access the raw data for the paper, but I was unable to locate any supporting code. Therefore, I was unable to replicate the analyses.

===PREPARING YOUR MANUSCRIPT===

===PREPARING YOUR REVISION IN SCHOLARONE===

Please ensure that you include a summary of your paper at Step 2 'Type, Title, & Abstract'. This should be no more than 100 words to explain to a non-scientific audience the key findings of your

research. This will be included in a weekly highlights email circulated by the Royal Society press office to national UK, international, and scientific news outlets to promote your work.

Author's Response to Decision Letter for (RSOS-201985.R0)

See Appendix B.

Decision letter (RSOS-201985.R1)

Dear Dr Meijer,

It is a pleasure to accept your manuscript entitled "Group housing and social dominance hierarchy affect circadian activity patterns in mice" in its current form for publication in Royal Society Open Science.

on behalf of Dr Oliver Schülke (Associate Editor) and Kevin Padian (Subject Editor)
openscience@royalsociety.org

Appendix A

Anita Kristiansen
Editorial Coordinator
Royal Society Open Science

Email: openscience@royalsociety.org

Leiden, October 30, 2020

Dear Dr. Kristiansen,

We thank you for the evaluation of our manuscript RSOS-192237. We thank you and the reviewers for their helpful comments, which we used to improve the manuscript.

Please find enclosed the revised manuscript and a letter detailing the changes made in accordance with the comments of the reviewers.

All authors have read the revised manuscript and approved submission. We sincerely hope that the clarifications and revisions will make the manuscript acceptable for publication.

Looking forward to hearing from you,

Sincerely,
Johanna Meijer

Prof. Dr. Johanna H. Meijer
Laboratory for Neurophysiology
Dept. Cell and Chemical Biology
Leiden University Medical Center
PO Box 9600 Mailbox S5-P
2300 RC Leiden, The Netherlands
Visiting address: Einthovenweg 20 Bldg 2, room T5-03
Mail: J.H.Meijer@LUMC.nl
Tel: 31 71 526 9760

Manuscript: RSOS-192237, revised

Title: Group housing and social dominance hierarchy affect circadian activity patterns in mice

Authors: Yuri Robbers, Mayke Tersteeg, Johanna H. Meijer and Claudia P. Coomans

We are grateful for the positive and constructive critiques, and greatly appreciate the opportunity to respond and revise the manuscript accordingly. All revisions are marked by track-changes in the text. In our letter our answers are in bold.

Associate Editor:

We would like to thank the associate editor for the very extensive review of our manuscript, which helped us to further improve the manuscript.

Introduction

The introduction after 90 words lands very quickly at lab mice without providing a broader introduction either to the behavioral ecology of the mouse, to the evolution of dominance hierarchies, to temporal niche segregation, or to density effects on grouping patterns as observed for example in wild cavies (the wild form of guinea pigs) and many other animals. This is especially relevant for this journal since it addresses a very broad readership.

We have now included a few sentences to the first paragraph of the introduction which broadens the significance of the manuscript. Furthermore, we have extended several other paragraphs in order to more clearly explain the background, and cited some additional literature.

P4, Line 29: Please provide a reference that established the term “Fujimoto effect”.

We have included the following reference to establish the term “Fujimoto effect”: Fujimoto K. Diurnal activity of mice in relation to social order. *Physiology and Ecology*. 1953;5:97-103.

P4, Line 33: Please explain what the crucial difference is between previous analyses and yours. What is circadian timescale? And consider here the work by one of your reviewers.

We have added a paragraph (page 3) and expanded another in order to explain these matters. The new material includes references to work by – among others – Oliver Schülke, Barbara König and James Curley.

P4, Line 39: Please explain what the costs and benefits of activity synchronization would be to provide some background and how these costs and benefits would change with density or grouping pattern. Most of the material presented on Lines 45ff needs to be moved to the methods and results section. It should be replaced by clear predictions derived from previous work and theory and a justification for the use of each response variable. The reader needs to be prepared for what is to come. Corticosterone measurements are mentioned in the last sentence but their use is not justified. Why is activity measured with four different activity indices derived from the RFID data – what is the added benefit and how do they relate to the research question and the Fujimoto effect? Why are they relevant here and what are they implied to measure? Why measure weight and what do you predict?

We have extensively rewritten the introduction paragraph and provided additional background to clarify our hypotheses.

Methods

The methods section lacks detail.

It is unclear how individual body mass was assessed.

We have added a sentence explaining this on page 5.

P5, Line 14: Please let the reader know whether 8 week old mice are adult.

We have added a sentence explaining that mice are considered adult at 8 weeks of age.

P5, Animals: The explanation of the design seems complicated because the replicates are labelled as experiments. There was only one type of treatment, i.e. to move group housed mice into individual cages and therefore it seems more straightforward to construe this as one experiment. The experiment consisted of observations of eight groups of six same-sex mice while house together and again after being moved to individual cages with half of the groups comprising only males and half only females.

We have rephrased the terminology as suggested.

P5, Line 35ff: The use of four measures of activity needs to be justified in the introduction.

We have expanded the paragraph in order to explain this.

P5, Line 44ff: It is unclear whether the data point that entered the Elo rating was a full conflict from the start to its end culminating in one mouse showing clear submission or the behavioral element. Listing all aggressive and submissive behaviors and then saying "Each of these behaviors was considered an interaction" suggest the latter which may inflate sample size for hierarchy construction and therefore inter-individual differences in winning success.

We have used only full sequences of such behavioural elements as interactions, and we have adapted the wording to make this clear.

P5, Line 60: Consider rephrasing. It was not the mice that were analyzed but their behavior.

We have rephrased this.

P6, Line 18: Please provide a reference that validates the EIA for the use with blood from mice.

We have included the following references: Kaikaew, et al. "Sex difference in corticosterone-induced insulin resistance in mice." *Endocrinology* 160, no. 10 (2019): 2367-2387; Ding, et al. "Late glucocorticoid receptor antagonism changes the outcome of adult life stress." *Psychoneuroendocrinology* 107 (2019): 169-178.

P6, Line 25: Please explain how you determined when the hierarchy was established and be very clear about whether you refer to the mice needing time or your tool needing time which is conceptually very different. If you refer to the burn-in period, there has been considerable progress in the development of the method in the past three years (see below).

The animals for each replication arrived as a group, and had been group housed together since birth. The hierarchy had therefore been established before our experiments started. The time referred to in the paper is strictly the time needed by our tool. We have clarified this in the text.

Please provide more detail on your Elo-rating including starting values and k values used as well as how you determined the burn in phase. Newton-Fisher et al 2017 (<https://doi.org/10.1007/s10764-017-9952-2>) have improved the Elo-rating method suggested by Albers & De Vries, Sanchez-Tojar et al. 2017 provide a guide for assessing the reliability of hierarchies constructed with different methods (DOI: 10.1111/1365-2656.12776), Goffe et al 2018 (DOI: 10.1111/2041-210X.13072) provide another objective method for determining the burn-in period, Vilette et al (<https://www.biorxiv.org/content/10.1101/692384v3>) for a guide for choosing the right ranking method for a particular research question.

Please provide the number of conflicts that entered the hierarchy analysis for each group as a quality control for the hierarchy.

This section has been expanded significantly. All these details have been added.

The description of the statistical methods used is too scarce to provide a clear picture. How did you deal with the dependency of data collected in the same group? It remains unclear whether data for one group (or one rank position) have been averaged before entering a test or whether a data point per individual was used in all analyses presented. Had samples sizes been reported with the test results, such confusion could have been avoided. Both approaches have their problems: if data were averaged per dominance rank before running correlations it is possible that a relationship was present in one or two groups but not the others. If the correlation was run on a sample of 4x6 individuals with 4 alphas and 4 betas and so forth, the biological dependencies introduced by housing the individuals in groups are ignored and samples size is unduly inflated. More description of the data are needed to judge whether hidden diversity between groups may have hampered the results based on averaged values.

The analyses and data have been described in more detail.

Which of your Wilcoxon rank-sum tests was a paired test and which was a test for independent samples and thus equivalent to a Mann-Whitney U-test. From my reading of the manuscript it seems that both types of tests have been used. Is it true that the test is referred to as a signed rank test if paired and as rank sum test if not?

We have changed the text and given the Mann-Whitney U-tests their proper name.

It is unclear which software has been used to run the tests and whether the adjustments required for both types of tests when run on small samples sizes have been used (Mundry & Fischer 1998 Anim Behav 56, 256-259).

We have used R. This has now been added to the text, with the following reference: R Core Team (2020). R: A language and environment for statistical computing. R Foundation for Statistical Computing, Vienna, Austria.

Results

Page 7, Line 10: It is unclear why and how a dominant individual or two dominant individuals have been determined for every group. Elo-rating provides a continuous measure of dominance without an objective threshold for determining who is dominant and who is subordinate.

If some analyses are run across ranks, why run comparisons between the alpha and the rest? Is there a good reason to expect dominance to not be linear but classifying? If so, this needs to be introduced early on and a method be devised for such classification. The description here makes it sound like this

was done by behavioral type (these mice were identified by their aggressive nature) and not by dominance rank or Elo-score. This is confusing.

There is specifically one point where the results show a clear difference between the two highest ranking males. We have highlighted that now and made it explicit that this is a purely an observation and we had no reason to expect this. Where relevant, we have replaced the test with a Kendall rank correlation and rephrased the text so that it is clear that we are looking at rank as being based upon a continuous measure.

P7, Line 22: It is unclear whether the interaction rate has been calculated per individual per hour or per group per hour.

The interaction rate has been calculated per individual per hour. This has now been clarified in the text.

P7, Line 27: It is unclear, how it was determined when a stable dominance hierarchy was established. See above for the conceptual difference between the animals establishing a stable hierarchy and our methods finding it.

This has been clarified in the text.

P7, Line 42: How was it established that the dominant mouse (is it not true that there were sometimes 2 of them) had a similar activity pattern to the other group members.

The activity patterns of the individual mice were assessed by comparing the actograms. This was done by experienced observers.

P7, Line 43: The sentence starting on line 43 has the same content as the one starting on line 48 and therefore is redundant and can be omitted.

The redundant sentence has been omitted.

P7, Line 49: The high value of W and the very low p value suggest that the test has been run on all 4x6 male mice in matched design which is problematic because of the non-independence of data point coming from the same group. Perhaps one could provide data per group in the supplementary material to double check whether the pattern is replicated in every group.

P7, Line 51: If the signed rank test is indeed the matched version of the Wilcoxon test, then it has not been applied correctly here for the comparison between dominants and subordinates which are two classes of independent data points. Why are the two top ranking males chosen here instead of either one or two in each group as suggested above was the pattern that emerged from the data. It seems that 4x2 dominants have been compared to 4x4 subordinates.

We thank the reviewer for this useful remark. We have now replaced the Wilcoxon test with a Kendall rank correlation.

P8, Line 11: The text describes a difference where the test was run as a correlation. This is problematic here because other tests have indeed compared dominants and subordinates.

This has been addressed, as in almost all cases the correlation is the better choice.

P8, Line 14: The correlation was run across all males from four groups. This is problematic because the ordinal rank variable contains a lot of ties then which needs correcting of Spearman's test and may even forbid it given that there are so many tied ranks. If it was not the ranks that entered the test but the Elo-Scores, they would need to be standardized before they can be used because Elo-Scores cannot be compared between groups. They are always relative to the other animals in the game. Some of the confusion about what was tested stems from the fact that figure 2 and 4 present data point per rank position only which suggest that this was what entered the tests.

We had used a correction of Spearman's test, but inspired by this comment we have decided it makes more sense to replace Spearman's rank correlation with Kendall's rank correlation, specifically the τ_b version, which was designed to deal well with tied ranks and low numbers of data points. We have also re-run the tests on standardized Elo scores, but are not convinced explaining the added layer of complexity to the reader is worth it in this case, as this makes no qualitative difference to the outcome.

P8, Line 5: Please specify what the values are (means across all males or medians).

This information has been added.

P9, Line 37: It may be revealing to run analyses on the change in CORT between ZT1 and ZT11 compared between housing conditions.

We had actually done this, but left it out of the paper due to space constraints. We have added it back in now.

P9, Weight: Why are all mice gaining body mass? Are they still growing when 8 weeks old? What is the biological background for this measure?

This is perfectly normal. We have explained this in the text now.

Discussion

Overall, the discussion reads like a list of posthoc explanations going for one response variable to the next. If the introduction would have set up the predictions from theory, one would now why CORT or body mass were measured and how this informs the research question. Here, one learns why CORT was measured after the entire activity results have been discussed. A more integrative treatise would be desirable.

We have rewritten the introduction and discussion extensively in order to address this comment.

P10, Line 26: In social behavior research the term emergent property is used like in system science, as a property that is brought about by the system that is more than its parts and that is more global than individual behavioral strategies can be. In that sense, the observations described here are not emergent properties but a naturally or sexually selected strategy to deal with increased density or with living in groups. I also do not understand the contrast with a possible reaction to a stressful situation as not being part of normal social biological behavior. The stress reaction is naturally selected and an integral part of the interaction of the individual with its environment.

The term emergent property has been dropped in favour of an evolutionary explanation.

P10, Line 34: Characteristics of the male and female dominance hierarchy have not been described in enough detail to say that they are similar. One could test linearity, steepness, relationship consistency, proportion of tied relationships or inconsistent relationships and the like.

We have performed additional analyses of the dominance hierarchies, i.e. the stability index, triangle transitivity and David's scores, and we have included the results in the paper. In all replications, the stability was high, there were no unknown or tied ranks, and all hierarchies were linear (significantly different from random).

P10, Line 46ff: This paragraph needs to be rewritten to include the papers suggested by the reviewer.

We have rewritten this paragraph extensively in order to accommodate the reviewer's comments.

P11, Line 3ff: Work on wild house mice suggests that male and female social organization differs markedly. Females may form close affiliative relationships with one or two other females, form a substructure to the broader grouping pattern and communally raise their young (work by Barbara Koenig from Zurich University and colleagues) whereas males establish local dominance hierarchies and regularly roam over larger areas seeking access to as many females as possible. Moreover, there is research showing sex differences in reaction to social stress cited on lines 57ff. It is therefore not surprising that male and female reactions to co-housing are different.

We have rewritten the introduction and discussion extensively in order to differences between male and female mice.

P11, Line 6: Sociability is usually used to describe an affiliative tendency, seeking rewarding social contact. In this sense, it is not straightforward why increased sociability would be a consequence of higher sociability of females. If the argument is that females are in close proximity more often because of their tendency to affiliate and that therefore the prima facie opportunity for aggression is also increased, the argument should be supported by data on proximity or affiliation.

We are merely hypothesizing, as we do not have any data on proximity or affiliation. We have rephrased the text to make this clear, and added some references to existing literature on sociability.

P11, Line 28: Here the strength of Elo-rating as a method that measured dominance on a continuous scale is highlighted as a strength. Why have so many analyses been run as comparisons between one or two dominants versus a seemingly homogenous group of subordinates? Is it intended to say that the big news is the occasional second dominant with a homogenous unstructured set of subordinates?

We have replaced most of those tests by correlations between dominants and subordinates.

P11, Line 33: What does it mean for two mice to have the highest dominance rating in a group? Did they have the same rating? Given how Elo-rating works, this is extremely unlikely.

It was an unnecessary and confusing statement. We have decided to take it out entirely.

P11, Line 57: What is stress itself? Stress is the phenomenon that the organism adjusts certain systems to maintain homeostasis in reaction to a stressor. Since the introduction failed to clearly spell out the predictions derived from theory, it must remain unclear here, why differences in CORT would be expected and in which way they would go. Also, CORT is now regarded more as a mediator of solutions to adaptive problems than an "indicator of stress" (<https://doi.org/10.1016/j.yhbeh.2016.07.001>).

We have rewritten this paragraph as well as the introduction to clarify this.

P12, Line 11ff: The crucial element lacking here seems to be mating competition. If males form dominance hierarchies mainly to regulate access to receptive females, dominant males may be more active as a part of their mate seeking behavior and maintenance of their dominance position.

The paragraph has been extended with mating competition.

P12, Line 29: Using activity pattern analysis (DFA) to measure the health outcome of changes in activity seems to be very indirect and perhaps even circular. Please adjust your wording here since the study has not quantified any measure of health.

We have rewritten this paragraph in order to address this comment.

References: The reference list is full of error concerning capitalization, omissions, superfluous material, and reference style.

We have corrected the reference list.

Figures: Text in the figures is very small in figures 1 and 3. It would also be very helpful to indicate in the figure in small text boxes in or above each panel instead of just numbering them a to h.

Figure legends: Since figures 1 and 3, or 2 and 4 show the same data quality for males and females the words males and females should be placed more towards the beginning of the title. As they stand now, one has to read three lines to get to the point where one is informed about whether one looks at male or female data in figure 2 and 4.

We have rewritten the captions slightly in order to solve this problem.

All the line figures need a measure of spread in the data to be included. Otherwise the reader cannot appreciate how similar or different animals in the four groups were and how much neighboring ranks overlap or not. To this end figures 2a and b and figures 4 a and b could be combined into one.

The measure of spread in the data has already been included in figure 2A and 4A. Figures 2B and 4B are merely meant to emphasize a pattern that may easily be missed in figure 2A and 2B. We had already made versions of 2B and 4B with measures of spread in them, but they obfuscated more than they revealed. That's why we left the spread out. We have changed the caption to more clearly explain the purpose of figure 2B and 4B.

Figure legend Fig 5: It is not true that "For male mice, but not for female mice, the CORT concentration is lower during single housing than compared to group housing." This applies only to ZT1 CORT but not ZT11 CORT. Please specify.

We have now specified this.

Reviewer 1:

This is a very interesting study that investigates the relationship between dominance rank and activity levels across the light cycle in male and female mice. The paper is noteworthy as the authors demonstrate that dominant males show increased daytime activity compared to subordinate mice, but

only when group housed. The authors also show that female mice form hierarchies with individual ranks, but that the patterns of activity are not different by rank. The authors use RFID and video methods to gather high quality data to support their findings.

I enjoyed reading the paper and believe it to be technically sound. I think it's a welcome addition to the literature. My major comments are however to do with some literature that the authors did not cite that actually supports their findings and conclusions. I feel that I (James Curley, UT Austin) should identify myself as the reviewer, as the majority of the papers I believe the authors should cite are from my own laboratory. I hope the authors and editor do not mind me doing this - I really am not in the habit of asking authors to cite my own work - but I feel that these papers share important findings that should be included.

We agree with the reviewer and have incorporated references at several points in the text to publications by Prof. Curley.

1. *In the introduction and discussion, the authors note papers that investigated how social group composition affects activity patterns, but they note that the relationship between dominance rank and temporal niche shifting was not investigated. To this end, the authors have not cited findings from two recent studies that need to be included. In one case, results from this study support their own findings.*

i) Lee et al 2018, Foraging dynamics are associated with social status and context in mouse social hierarchies, PeerJ. <https://www.ncbi.nlm.nih.gov/pubmed/30258716> . This paper demonstrates that dominant male mice living in hierarchies show consistently higher levels of activity and feeding/drinking throughout the 24hour period, including during the light period, compared to subordinate mice. These findings are congruent with the current study's findings that dominant mice increase their daytime activity when group housed.

The proposed reason for the increased daytime activity by dominant mice in the Lee et al paper is that this is due to increased behavioral (patrolling and territorial defense) and energetic demands of being dominant - as argued on line 12 of page 12.

ii) Karamihalev et al 2019, Social context and dominance status contribute to sleep patterns and quality in groups of freely-moving mice, Sci Rep. <https://www.nature.com/articles/s41598-019-51375-7> This paper shows that dominance status influences sleep quality.

Both of these papers speak importantly to the issues and questions raised in the current paper and should be included.

We agree with the reviewer and have incorporated references in the text to these two publications.

2. *The findings that females form social hierarchies. This is a very important finding as so few studies have investigated whether female mice form hierarchies. This confirms findings in Williamson et al 2019 Sci Reports <https://www.nature.com/articles/s41598-019-43747-w>, that females can indeed form hierarchies. In that paper we also review the few other studies that have investigated that question. The current study has excellent quality data that provide strong support for the hypothesis that female mice do indeed form hierarchies.*

In the revised manuscript we have included this reference on page 13.

3. *In the discussion (page 11 lines 13-38), the authors describe the type of hierarchies that mice form. They discuss how they are usually described as having one dominant and several subordinate*

animals. They state that their ELO rating method enables them to identify individual ranks. I would argue that the reason why previous studies reported one dominant animal and several subordinate is largely due to the type of data that was collected meaning those authors were unable to identify the stratification of ranks. In our own work (e.g. see Williamson et al 2016 "Temporal dynamics of social hierarchy formation and maintenance in male mice" <https://www.sciencedirect.com/science/article/pii/S0003347216000798> we applied a Glicko method (essentially an improved version of ELO) that we used to identify individual ranks in mice. It should be mentioned that current studies of social hierarchies in mice are revealing that mice do form hierarchies with individual ranks being identifiable. Further, the use of ELO is not actually that strong a method for identifying clearly different rank orders. To do that, you should technically use I&SI ranking (although this does correlate well with ELO ranking). We have used I&SI rankings in a series of papers to show that mice do indeed form hierarchies with individual ranks. Although in the current study, I think that ELO is sufficient. All this is to say, that I believe the authors excellent high quality data is confirming these findings, which is very welcome, and suggests that the older literature were assuming a pattern of dominance relationships without being able to detect the more detailed gradations in ranks.

We thank the reviewer for this comment and have included now a sentence to refer to the Glicko ranking.

Reviewer 2:

Robbers et al. report the impact of group housing and dominance on the daily activity of C57BL/6 mice. It is highly interesting how individual species adapt their activity pattern to their social environment, and relatively little is known about this, both in the lab and in the field. I therefore like this study; the approach is very basic and the outcome can and should be published.

Generally, I do not have many comments on the manuscript. The paper is well written and to the point. The only, minor comment – or actually question – I have, is on the analysis - I wonder why the authors did not choose to use non-parametric models with individuals nested in their social groups? I however realize that the approach as chosen now should be more conservative, and probably does not differ much for for example the near significant overall difference in activity of group- and single housed females.

I have no further comments; please check symbol indications in figure descriptions

We thank the reviewer for his/her kind words. With regards to the analysis, we have decided against the non-parametric model with individuals nested in their social groups because we only have four all-male and four all-female groups consisting of four individuals each. For the most logical choice of non-parametric model, the adjusted Friedman test for the nested design¹ that is not enough data. We hope this paper inspires further experiments that will include sufficient replications to run more complex statistical analyses of the kind proposed by the reviewer.

¹ S. J. M. Brits & H. H. Lemmer (1990) *An adjusted Friedman test for the nested design*, Communications in Statistics - Theory and Methods, 19:5, 1837-1855, DOI: [10.1080/03610929008830294](https://doi.org/10.1080/03610929008830294)

Appendix B

Oliver Schülke
Associate Editor
Royal Society Open Science

Email: openscience@royalsociety.org

Leiden, December 13, 2020

Dear Dr. Schülke,

We thank you for the evaluation of our manuscript RSOS-192237.

We have added the R code to Dryad, thereby allowing for replication of our analyses.

Looking forward to hearing from you,

Sincerely,
Johanna Meijer

Prof. Dr. Johanna H. Meijer
Laboratory for Neurophysiology
Dept. Cell and Chemical Biology
Leiden University Medical Center
PO Box 9600 Mailbox S5-P
2300 RC Leiden, The Netherlands
Visiting address: Einthovenweg 20 Bldg 2, room T5-03
Mail: J.H.Meijer@LUMC.nl
Tel: 31 71 526 9760